# Creating artificial human genomes using generative neural networks

Burak Yelmen [1,2,3]*, Aurélien Decelle [3,4], Linda Ongaro [1,2], Davide Marnetto [1], Corentin Tallec [3], Francesco Montinaro [1,5], Cyril Furtlehner [3], Luca Pagani [1,6], Flora Jay [3]*

1 Institute of Genomics, University of Tartu, Tartu, Estonia, 2 Institute of Molecular and Cell Biology, University of Tartu, Tartu, Estonia, 3 Laboratoire de Recherche en Informatique, CNRS UMR 8623, Université Paris-Sud, Université Paris-Saclay, Paris, France, 4 Departamento de Física Téorica I, Universidad Complutense, Madrid, Spain, 5 Department of Biology-Genetics, University of Bari, Bari, Italy, 6 APE Lab, Department of Biology, University of Padova, Padova, Italy

* burakyelmen@gmail.com (BY); flora.jay@lri.fr (FJ)

**Data Availability Statement:** The authors confirm that all data associated with the study are fully available without restriction. Training data was obtained from literature and can be accessed through "https://genomics.ut.ee/en/biobank.ee/

## Abstract

Generative models have shown breakthroughs in a wide spectrum of domains due to recent advancements in machine learning algorithms and increased computational power. Despite these impressive achievements, the ability of generative models to create realistic synthetic data is still under-exploited in genetics and absent from population genetics. Yet a known limitation in the field is the reduced access to many genetic databases due to concerns about violations of individual privacy, although they would provide a rich resource for data mining and integration towards advancing genetic studies. In this study, we demonstrated that deep generative adversarial networks (GANs) and restricted Boltzmann machines (RBMs) can be trained to learn the complex distributions of real genomic datasets and generate novel high-quality artificial genomes (AGs) with none to little privacy loss. We show that our generated AGs replicate characteristics of the source dataset such as allele frequencies, linkage disequilibrium, pairwise haplotype distances and population structure. Moreover, they can also inherit complex features such as signals of selection. To illustrate the promising outcomes of our method, we showed that imputation quality for low frequency alleles can be improved by data augmentation to reference panels with AGs and that the RBM latent space provides a relevant encoding of the data, hence allowing further exploration of the reference dataset and features for solving supervised tasks. Generative models and AGs have the potential to become valuable assets in genetic studies by providing a rich yet compact representation of existing genomes and high-quality, easy-access and anonymous alternatives for private databases.

## Author summary

Generative neural networks have been effectively used in many different domains in the last decade, including machine dreamt photo-realistic imagery. In our work, we apply a similar concept to genetic data to automatically learn its structure and, for the first time,

data-access" and "https://www.
internationalgenome.org/data". Relevant code and
information can be accessed from "https://gitlab.
inria.fr/ml_genetics/public/artificial_genomes".

**Funding:** This work was supported by the
European Union through the European Regional
Development Fund (Project No. 2014-
2020.4.01.16-0024, MOBTT53: LP, DM, BY;
Project No. 2014-2020.4.01.16-0030: LO, FM); the
Estonian Research Council grant PUT (PRG243):
LP; DIM One Health 2017 (number
RPH17094JJP): FJ; Comunidad de Madrid and the
Complutense University of Madrid (Spain) through
the Atracción de Talento program (Ref. 2019-T1/
TIC-13298): AD; Laboratoire de Recherche en
Informatique "Promoting Collaborations &
Scientific Excellence of Young Researchers": FJ.
The funders had no role in study design, data
collection and analysis, decision to publish, or
preparation of the manuscript.

**Competing interests:** The authors have declared
that no competing interests exist.

produce high quality realistic genomes. These novel genomes are distinct from the original ones used for training the generative networks. We show that artificial genomes, as we name them, retain many complex characteristics of real genomes and the heterogeneous relationships between individuals. They can be used in intricate analyses such as imputation of missing data as we demonstrated. We believe they have a high potential to become alternatives for many genome databases which are not publicly available or require long application procedures or collaborations and remove an important accessibility barrier in genomic research in particular for underrepresented populations.

## Introduction

Availability of genetic data has increased tremendously due to advances in sequencing technologies and reduced costs [1]. The vast amount of human genetic data is used in a wide range of fields, from medicine to evolution. Despite the advances, cost is still a limiting factor and more data is always welcome, especially in population genetics and genome-wide association studies (GWAS) which usually require substantial amounts of samples. Partially related to the costs but also to the research bias toward studying populations of European ancestry, many autochthonous populations are under-represented in genetic databases, diminishing the extent of the resolution in many studies [2–5]. Additionally, the majority of the data held by government institutions and private companies is considered sensitive and not easily accessible due to privacy issues, exhibiting yet another barrier for scientific work. A class of machine learning methods called generative models might provide a suitable solution to these problems.

Generative models are used in unsupervised machine learning to discover intrinsic properties of data and produce new data points based on those. In the last decade, generative models have been studied and applied in many domains of machine learning [6–8]. There have also been a few applications in the genetics field [9–12], one specific study focusing on generating DNA sequences via deep generative models to capture protein binding properties [13]. Among the various generative approaches, we focus on two of them in this study, generative adversarial networks (GANs) and restricted Boltzmann machines (RBMs). GANs are generative neural networks which are capable of learning complex data distributions in a variety of domains [14]. A GAN consists of two neural networks, a generator and a discriminator, which compete in a zero-sum game (S1 Fig). During training, the generator produces new instances while the discriminator evaluates their authenticity. The training objective consists in learning the data distribution in a way such that the new instances created by the generator cannot be distinguished from true data by the discriminator. Since their first introduction, there have been several successful applications of GANs, ranging from generating high quality realistic imagery to gap filling in texts [15,16]. GANs are currently the state-of-the-art models for generating realistic images [17].

A restricted Boltzmann machine, initially called Harmonium, is another generative model which is a type of neural network capable of learning probability distributions through input data [18,19]. RBMs are two-layer neural networks consisting of an input (visible) layer and a hidden layer (S2 Fig). The learning procedure for the RBM consists in maximizing the likelihood function over the visible variables of the model. This procedure is done by adjusting the weights such that the correlations between the visible and hidden variables on both the dataset and sampled configurations from the RBM converge. Then RBM models recreate data in an unsupervised manner through many forward and backward passes between these two layers (Gibbs sampling), corresponding to sampling from the learned distribution. The output of the

hidden layer goes through an activation function, which in return becomes the input for the hidden layer. Although mostly overshadowed by recently introduced approaches such as GANs or Variational Autoencoders [20], RBMs have been used effectively for different tasks (such as collaborative filtering for recommender systems, image or document classification) and are the main components of deep belief networks [21–23].

Here we propose and compare a prototype GAN model along with an RBM model to create Artificial Genomes (AGs) which can mimic real genomes and capture population structure along with other characteristics of real genomes. These prototypes are compared to alternative generative models based on multiple summaries of the data and we explore whether a meaningful encoding of real data has been learned. Finally, we investigate the potential of using AGs as proxies for private datasets that are not accessible in order to address various genomic tasks such as imputation or selection scans.

## Results

### Reconstructing genome-wide population structure

Initially we created AGs with GAN, RBM, and two simple generative models for comparison: a Bernoulli and a Markov chain model (see Materials & Methods) using 2504 individuals (5008 haplotypes) from 1000 Genomes data [24], spanning 805 SNPs from all chromosomes which reflect a high proportion of the population structure present in the whole dataset [25]. Both GAN and RBM models capture a good portion of the population structure present in 1000 Genomes data while the other two models could only produce instances centered around 0 on principal component analysis (PCA) space (Fig 1). All major modes, corresponding to African, European and Asian genomes, are well represented in AGs produced by GAN and RBM models and absent for the Markovian and Bernouilli models. Wasserstein distances between the 2D PCA representations of real versus generated individuals were closer to 0 for GAN (0.006), RBM (0.006) and the test set (0.01) than for the Markovian (0.124) and Bernoulli (0.240) models. Uniform manifold approximation and projection (UMAP) mapping results (performed on the combined dataset) lead to similar conclusions (Wasserstein 2D distance from real for GAN: 0.021, RBM: 0.091, Markovian: 0.088, Bernoulli: 0.127) although the RBM distribution is slightly shifted compare to the real one (S3 Fig). We additionally computed the distribution of pairwise differences of haploid genomes within a single dataset or between the real and artificial datasets (S4 Fig). The real, GAN and RBM distributions present similar multimodal patterns reflecting the underlying population structure (in particular the RBM distribution exhibits three modes corresponding to the average distances between European and Asian, European and African, or African and Asian populations. The overall real pairwise distribution is captured as accurately by the GAN (Wasserstein distance between real and generated distributions: 3.24) and RBM (6.21) models than by a test set (5.06) and those clearly outperform the binomial (42.20) and Markovian (37.92) models. No real genome was copied into the AGs (0 identical pair). Since GANs and RBMs showed an excellent performance for this use case, we further explored other characteristics using only these two models.

### Reconstructing local high-density haplotype structure

To evaluate if high quality artificial dense genome sequences can also be created by generative models, we applied the GAN and RBM methods to a 10K SNP region using (i) the same individuals from 1000 Genomes data and (ii) 1000 Estonian individuals from the high coverage Estonian Biobank [26] to generate artificial genomes. PCA results of AGs spanning consecutive 10K SNPs show that both GAN and RBM models can still capture the relatively toned-down population structure (S5 Fig; 2D Wasserstein distances for 1000 Genomes and Estonian

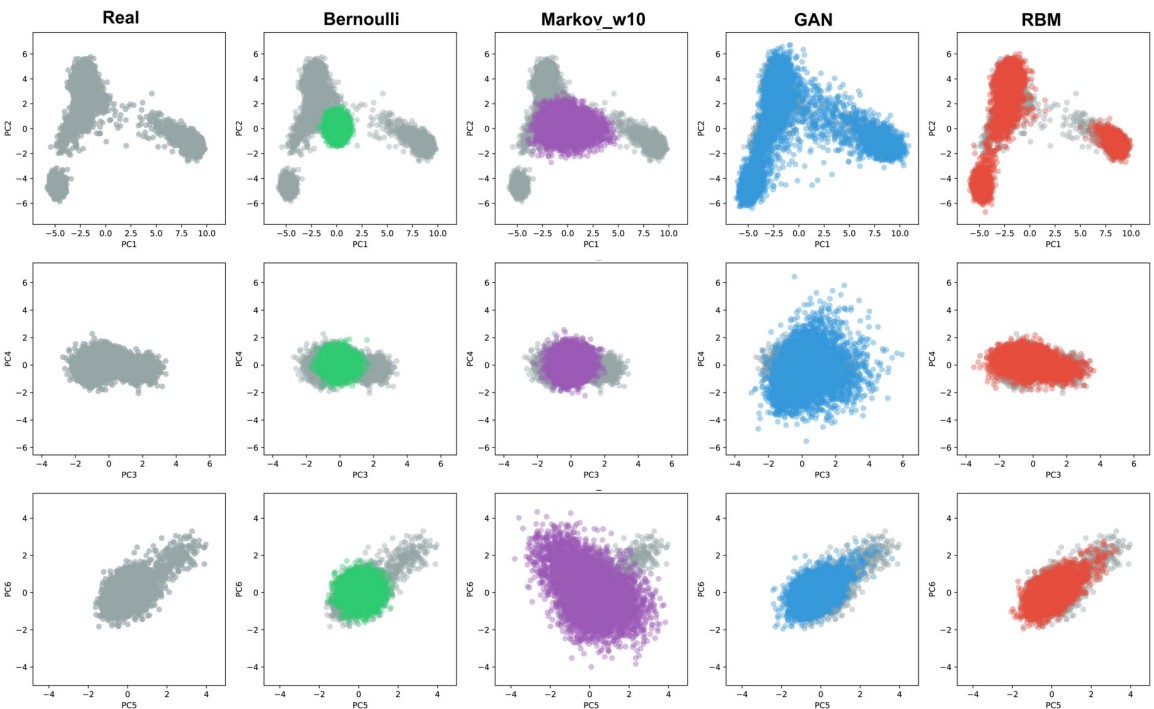

**Fig 1. The six first axes of a single PCA applied to real (gray) and artificial genomes (AGs) generated via Bernoulli (green), Markov chain (purple), GAN (blue) and RBM (red) models.** There are 5000 haplotypes for each AG dataset and 5008 (2504 genomes) for the real dataset from 1000 Genomes spanning 805 informative SNPs. See Materials & Methods for detailed explanation of the generation procedures.

respectively: 0.004 and 0.011 for GAN, 0.011 and 0.006 for RBM, 0.004 and 0.002 for test sets) as well as the overall distribution of pairwise distances (S6 Fig). Looking at the allele frequency comparison between real and artificial genomes, we see that especially GAN performs poorly for low frequency alleles, due to a lack of representation of these alleles in the AGs whereas RBM seems to have wider distribution over all frequencies (S7 Fig; correlation between real and generated for 1000 Genomes and Estonian respectively: 0.99 and 0.91 for GAN, 0.94 and 0.94 for RBM, 0.99 and 0.99 for test sets). The overall pairwise distributions are fitted better by the RBM than the GAN (Wasserstein distance 117 and 227 for GAN, 38 and 26 for RBM, 22 and 16 for test sets). On the other hand, the distribution of the distance of real genomes to the closest AG neighbour shows that GAN model, although slightly underfitting, outperforms RBM model, for which an excess of small distances points towards overfitting, visible by the distribution being closer to the zero point (S8 Fig).

Additionally, we performed linkage disequilibrium (LD) analyses comparing artificial and real genomes to assess how successfully the AGs imitate short and long range correlations between SNPs. Pairwise LD matrices for real and artificial genomes all show a similar block pattern demonstrating that GAN and RBM accurately captured the overall structure with SNPs having higher linkage in specific regions (Fig 2A). However, plotting LD as a function of the SNP distance showed that all models capture weaker correlation, with RBM outperforming the GAN model perhaps due to its slightly overfitting characteristic (Fig 2B). However, correlations between real and generated LD across all pairs were similar for GAN and RBM (for 1000 Genomes and Estonian respectively: 0.95 and 0.97 for GAN, 0.94 and 0.98 for RBM) and slightly lower than for test sets (0.99 and 1.0) (S9 Fig). LD can be seen as a two-point correlation statistic, we also investigated 3-point correlation statistics, that represent the amount of

a.

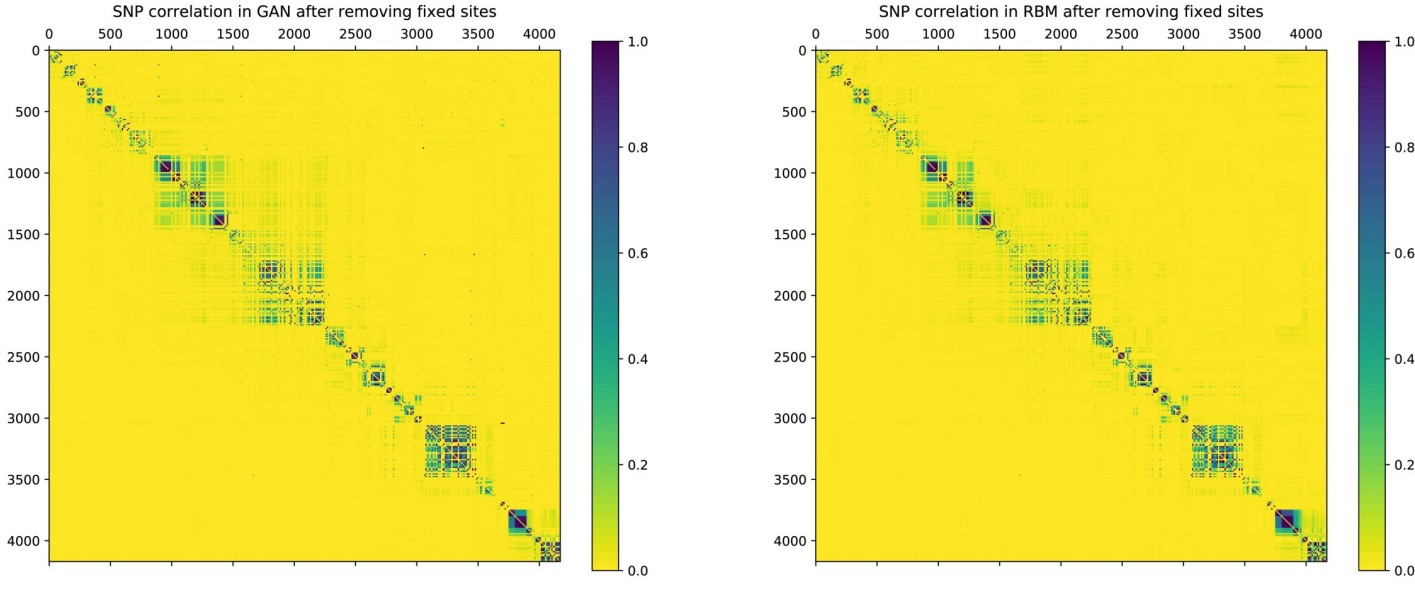

b.

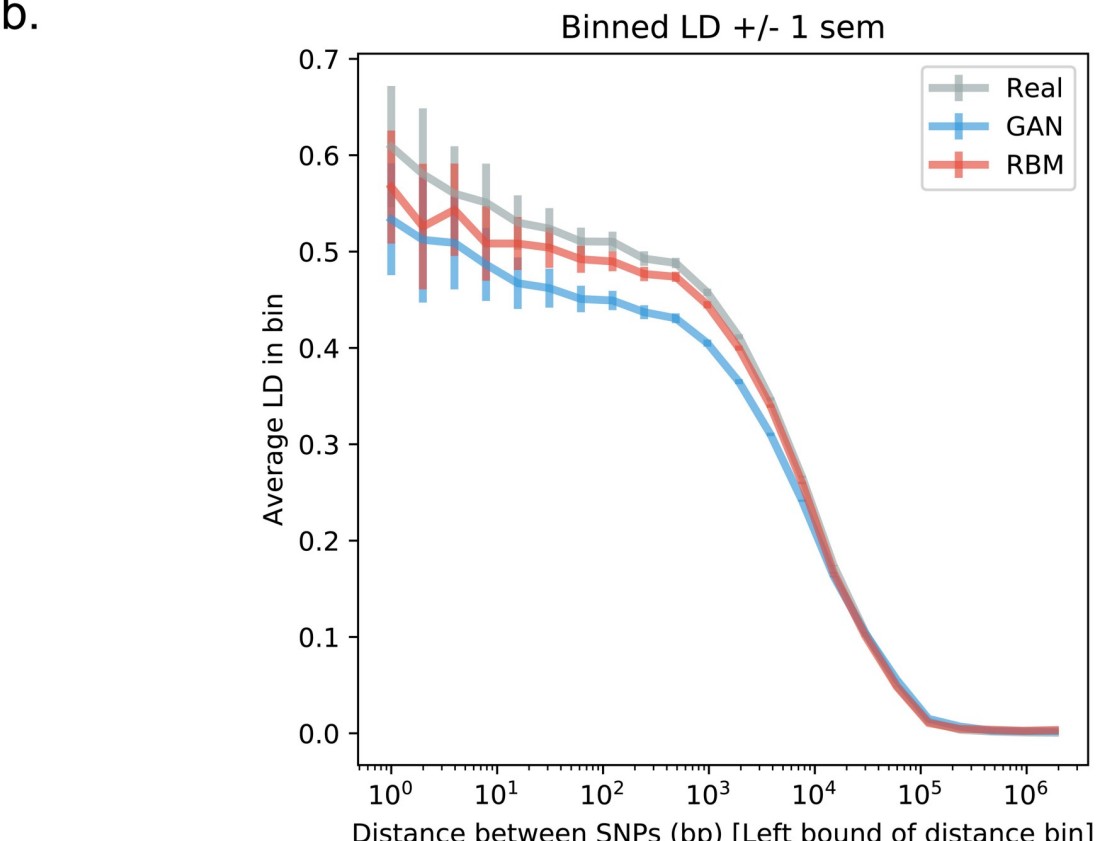

**Fig 2. Linkage disequilibrium (LD) analysis on real and artificial Estonian genomes. a)** Correlation ($r^2$) matrices of SNPs. Lower triangular parts are SNP pairwise correlation in real genomes and upper triangular parts are SNP pairwise correlation in artificial genomes. **b)** LD as a function of SNP distance after removing sites that are fixed in at least in one dataset. Pairwise SNP distances were stratified into 50 bins and for each distance bin, the correlation was averaged over all pairs of SNPs belonging to the bin.

correlation between triplets of SNPs and thus characterize more complex correlation patterns in datasets (S10 Fig). To further determine the haplotypic integrity of AGs, we performed ChromoPainter [27] and Haplostrips [28] analyses on AGs created using Estonians as the training data. We did not observe separate clustering of real and artificial genomes with Haplostrips (S11 Fig). However, the majority of the AGs produced via GAN model displayed an excess of short chunks when painted against 1000 Genomes individuals, whereas we do not observe this for RBM AGs (S12 Fig). Average European chunk length over 100 individuals for GAN AGs was 44.21%, RBM AGs was 54.92%, whereas for real Estonian genomes, it was 62.83%.

After demonstrating that our models generated realistic AGs according to the described summary statistics, we investigated further whether they respected privacy by measuring the extent of overfitting. We calculated two metrics of resemblance and privacy, the nearest neighbour adversarial accuracy ($AA_{TS}$) and privacy loss presented in a recent study [29]. $AA_{TS}$ score measures whether two datasets were generated by the same distribution based on the distances between all data points and their nearest neighbours in each set. When applied to artificial and real datasets, a score between 0.5 and 1 indicates underfitting, between 0 and 0.5 overfitting (and likely privacy loss), and exactly 0.5 indicates that the datasets are indistinguishable. By using an additional real test set, it is also possible to calculate a privacy loss score that is positive in case of information leakage, negative otherwise, and approximately ranges from -0.5 to 0.5. Computed on our generated data, both scores support haplotypic pairwise difference results confirming low privacy loss for GAN AGs with a score similar to an independent Estonian test set never used during training (GAN: 0.027; Test: 0.021) and the overfitting nature of RBM AGs with a high risk of privacy leakage (RBM privacy loss: 0.327; S13 Fig). Using an alternative sampling scheme for the RBM (see Material and Methods) slightly reduced privacy loss (restrained under 0.2 for low number of epochs; S14 Fig). A dataset produced by this alternative scheme had only a slight decrease in quality of the summary statistics while the privacy loss was reduced to 0.1. For this scheme, the correlation between generated and true allele frequencies was 0.92 (instead of 0.95 for the previous RBM and 0.98 for GAN), the correlation for LD values was 0.97 (RBM:0.98, GAN:0.97), the 2D-Wasserstein distance for the PCA representations was 0.026 (RBM: 0.006, GAN: 0.011, RBM sampling initialized randomly: 0.339), the Wasserstein distance for the pairwise distribution was 97 (RBM: 26, GAN: 227, RBM sampling initialized randomly: 689).

## Selection tests

We additionally performed cross population extended haplotype homozygosity (XP-EHH) and population branch statistic (PBS) on a 3348 SNP region homogenously dispersed over chromosome 15 to assess if AGs can also capture selection signals. Both XP-EHH and PBS results provided high correlation between the scores of real and artificial genomes (Fig 3). Similar peaks were observed for real and artificial genome datasets (see Discussion).

## Imputation

Since it has been shown in previous studies that imputation scores can be improved using additional population specific reference panels [30,31], as a possible future use case, we tried imputing real Estonian genomes using 1000 Genomes reference panel and additional artificial reference panels with Impute2 software [32]. Both combined RBM AG and combined GAN AG panels outperformed 1000 Genomes panel for the lowest MAF bin (for MAF < 0.05, 2.5% and 4.4% improvement respectively) which had 5926 SNPs out of 9230 total (Fig 4). Also mean info metric over all SNPs were intermediate between the regular imputation scheme

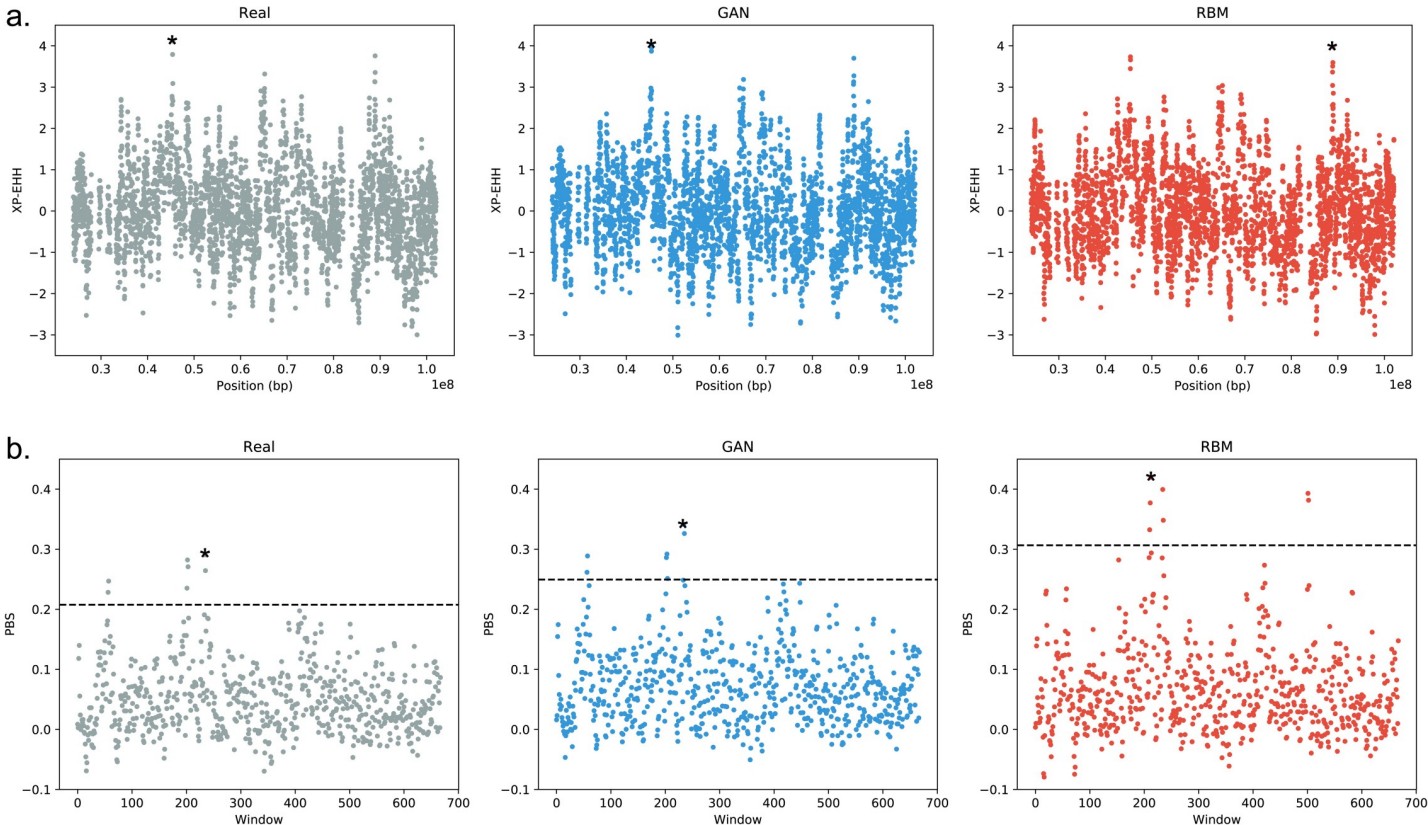

**Fig 3. Selection tests on chromosome 15. a)** Standardized XP-EHH scores of real and artificial Estonian genomes using 1000 Genomes Yoruba population (YRI) as the complementary population. Correlation coefficient between real and GAN XP-EHH score is 0.902, between real and RBM XP-EHH score is 0.887. **b)** PBS scores of real and artificial Estonian genomes using 1000 Genomes Yoruba (YRI) and Japanese (JPT) populations as the complementary populations. PBS window size is 10 and step size is 5. Dotted black line corresponds to the 99th percentile. Correlation coefficient between real and GAN PBS score is 0.923, between real and RBM PBS score is 0.755. Highest peaks are marked by an asterisk.

(1000 Genomes panel only) and the 'perfect' scheme (panel including private Estonian samples). The scores were 1.3%, 2.3%, and 6.9% higher for the combined RBM, GAN and real Estonian panels respectively, compared to the panel with only 1000 Genomes samples. However, aside from the lowest MAF bin, 1000 Genomes panel slightly outperformed both concatenated AG panels for all the higher bins (by 0.05% to 0.6% depending on the frequency bin). This might be a manifestation of haplotypic deformities in AGs that might have disrupted the imputation algorithm.

## Data encoding and visualization via RBM model

Furthermore, similarly to tSNE and UMAP, RBMs perform a non-linear dimension reduction of the data and provides a suitable representation of a genomic dataset as a by-product based on the non-linear feature space associated to the hidden layer (Materials & Methods). As Diaz-Papkovich et al [33], we found that the RBM representation differs from the linear PCA ones (S15 Fig), although the general structure identified by the two lower rank components is highly similar. Like in a PCA, African, East Asian, and to a lesser extent, European populations stand out on the two first components yet the relative positions differ slightly from PCA to RBM. In particular, the Finnish appear slightly more isolated from the other European populations on the first component of the RBM. South Asians are located at the center separated from

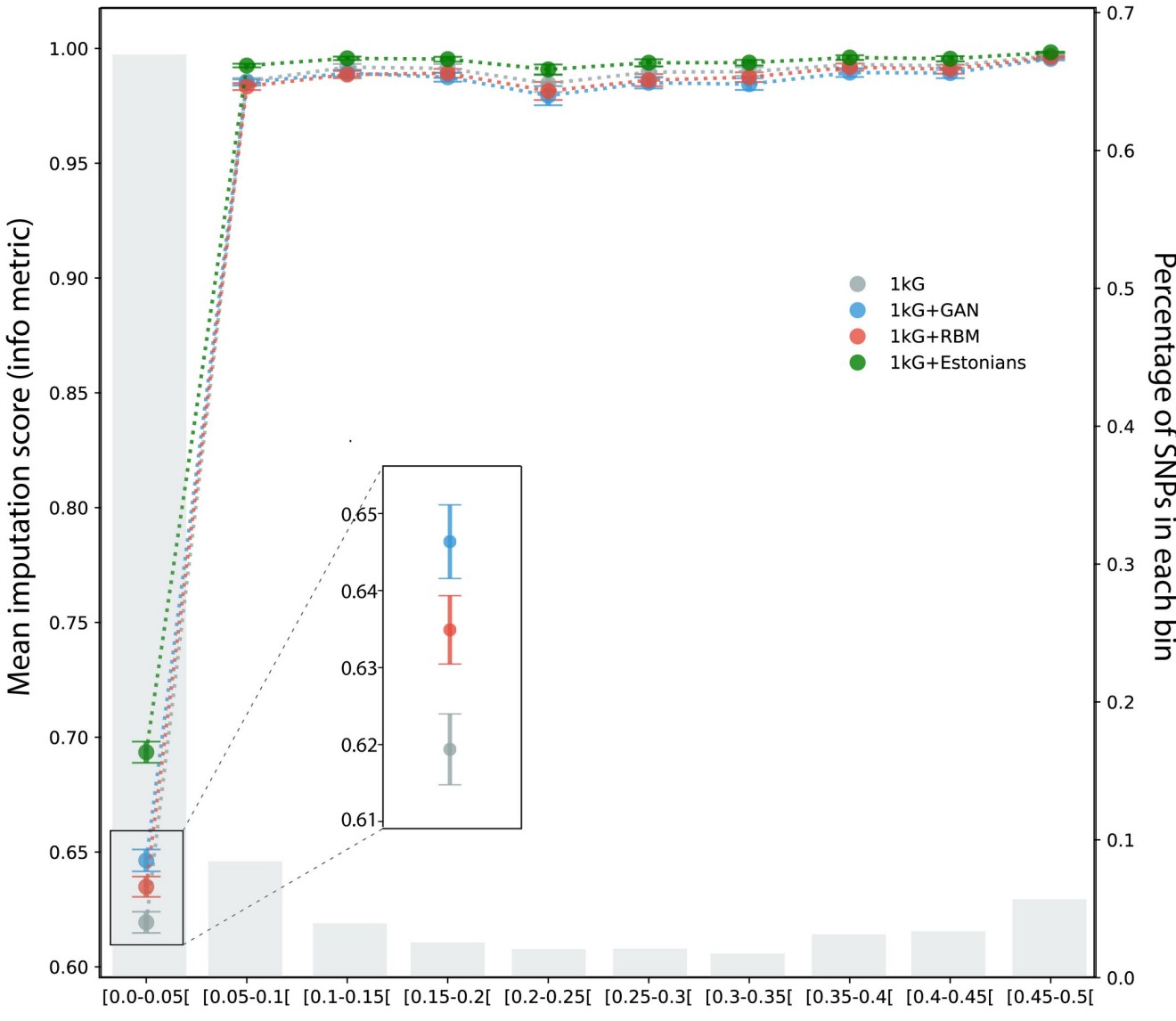

**Fig 4. Imputation evaluation of three different reference panels based on Impute2 software's info metric.** Imputation was performed on 8678 Estonian individuals (which were not used in training of GAN and RBM models) using only 1000 Genomes panel (gray), combined 1000 Genomes and Estonian genomes used in training (green), combined 1000 Genomes and GAN artificial genomes panel (blue) and combined 1000 Genomes and RBM artificial genomes panel (red). SNPs were divided into 10 MAF bins, from 0.05 to 0.5, after which mean info metric values were calculated. Grey bars show the percentage of SNPs which belong to each bin.

Europeans, partially overlapping with American populations, and stand out at dimension 5 and higher (versus 3 for the PCA). The third RBM component exhibit a stronger gradient than PCA for Peruvian and Mexican individuals and might reflect their gradient in Native American ancestry. Finally, RBM still exhibits population structure in components higher than 7, contrary to PCA. Interestingly when screening the hidden node activations, we observed that different populations or groups activate different hidden nodes, each one representing a specific combination of SNPs, thereby confirming that the hidden layer provides a meaningful encoding of the data (S16 Fig).

## Comparison with alternative generative models

We additionally performed tests to compare AGs with advanced methods used to generate genomes. One such method is the copying model [34] implemented in HAPGEN2 [35]. Although genomes generated via this approach performed very well in terms of SFS, LD and PCA, there was extensive overfitting and privacy loss and multiple individuals (747 identical haplotypes) were directly copied from the original dataset (S17 Fig).

Another commonly used approach to generate genomes is coalescent simulations. Although it is inherently difficult to make a fair comparison since coalescent simulations require additional (demographic) parameters and do not provide the desired one-to-one SNP correspondence (see Discussion), we compared SFS and LD decay of AGs with genomes simulated via a previously inferred demographic model [36] using HapMapII genetic map [37] implemented in stdpopsim [38–41]. Initially, we performed PCA and checked the allele frequency distribution compared to real genomes (S18 Fig). The reasoning behind PCA was to demonstrate that coalescent simulation genomes cannot be combined with real genomes since they exist in different planes. Since we had selected SNPs for 1000 Genomes and Estonian datasets to be overlapping, we removed alleles below 0.1 frequency from all datasets to eliminate biases and analyzed LD decay and allele frequency spectrum (S19 Fig). For both summary statistics, coalescent simulation genomes performed well. Still, direct comparison of frequencies SNPs by SNPs, LD pairs by pairs, PCA, $AA_{TS}$ or distributions of pairwise distances between real and generated data are not feasible for coalescent simulations. Notably, the demographic model we adopted was optimized for another European population (CEU from the 1000 Genomes Project), since an in depth study of the demographic properties of Estonians, our target population, required extensive efforts beyond the scope of this paper and in themselves a cost to be considered when adopting coalescent simulations as a generative model.

## Discussion

In this study, we applied generative models to produce artificial genomes and evaluated their characteristics. To the best of our knowledge, this is the first application of GAN and RBM models in the population genetics context, displaying an overall promising applicability. We showed that population structure and frequency-based features of real populations can successfully be preserved in AGs created using GAN and RBM models. Furthermore, both models can be applied to sparse or dense SNP data given a large enough number of training individuals. Our different trials showed that the minimum required number of individuals for training is highly variable (i.e. to avoid training failures such as mode collapse or incomplete training without converging to an ideal mode) depending on the unknown dimension of the dataset, which is linked to the type of data to be generated and the population(s). Since haplotype data is more informative, we created haplotypes for the analyses but we also demonstrated that the GAN model can be applied to genotype data too, by simply combining two haplotypes if the training data is not phased (see Materials & Methods). In addition, we showed that it might be possible to generate AGs with simple phenotypic traits through genotype data (see S1 Table and S1 Text). Even though there were only two simple classes, blue and brown eye color phenotypes, generative models can be improved in the future to hold the capability to produce artificial datasets combining AGs with multiple phenotypes

One major drawback of the proposed models is that, due to computational limitations, they cannot yet be harnessed to create whole artificial genomes but rather snippets or sequential dense chunks. It should be possible to create whole genomes by independently training and generating multiple chunks from different genomic regions using a single uniform population such as Estonians and stitching them together to create a longer, genome-like, sequence for

each AG individual. To mitigate possible disruptions in the long-range haplotype structure, these chunks can be defined based on "approximately independent LD blocks" [42]. Yet for data with population structure, it would be difficult to decide which combination of chunks can form a single genome. Statistics such as FST or generative models conditioned on group labels might be utilized to overcome this issue. On the other hand, a collection of chunks covering the whole genome can be used for analyses based solely on allele frequencies without any stitching. A technically different approach would be to adapt convolutional GANs for AG generation [43].

Another problem arose due to rare alleles, especially for the GAN model. We showed that nearly half of the alleles become fixed in the GAN AGs in the 10K SNP dataset, whereas RBM AGs capture more of the rare alleles present in real genomes (S20 Fig). A known issue in GAN training is mode collapse [44], which occurs when the generator fails to cover the full support of the data distribution. This type of failure could explain the inability of GANs to generate rare alleles. For some applications that depend on rare alleles, GAN models less sensitive to mode dropping may be a promising alternative [45–47].

An important use case for the AGs in the future might be creating public versions of private genome banks. Through enhancements in scientific and technology knowledge, genetic data becomes more and more sensitive in terms of privacy. AGs might offer a versatile solution to this delicate issue in the future, protecting the anonymity of real individuals. They can be utilized as input for downstream operations such as forward steps of a specific evolutionary process for which they can become variations of the real datasets (similar to bootstrap), or they can be the sole input when the real dataset is not accessible. Initializing a simulation with real data is a procedure that is commonly used in population genetics [48,49]. Our results showed that GAN AGs are possibly underfitting while, on the contrary, RBM AGs are overfitting, based on distribution of minimum distance to the closest neighbour (S8 Fig) and $AA_{TS}$ scores (S13A Fig), although we showed how overfitting could be restrained by integrating $AA_{TS}$ scores within our models as a criterion for early stopping in training (before the networks start overfitting) and by modifying the RBM sampling scheme. In the context of the privacy issue, GAN AGs have a slight advantage since underfitting and low leakage information is preferable. More distant AGs would hypothetically be harder to be traced back to the original genomes. We also tested the sensitivity of the $AA_{TS}$ score and privacy loss (S21 Fig). It appears that both scores are affected very slightly when we add only a few real genomes to the AG dataset from the training set. Although this case is easily detectable by examining the extreme left tail of the pairwise distribution, it advocates for combining multiple privacy loss criteria and developing other sensitive measurement techniques for better assessment of generated AGs. Additionally, even though we did not detect exact copies of real genomes in AG sets created either by RBM or GAN models, it is a very complicated task to determine if the generated samples can be traced back to the originals. Reliable measurements need to be developed in the future to assure complete anonymity of the source individuals given the released AGs. In particular, we will investigate whether the differential privacy framework is performant in the context of large population genomics datasets [50,51].

Imputation results demonstrated promising outcomes especially for population specific low frequency alleles. However, imputation with both RBM and GAN AGs integrated reference panels showed slight decrease of info metric for higher frequency alleles compared to only 1000 Genomes panel (Fig 4). Increasing the number of AGs did not affect the results significantly. We initially speculated that this decrease might be related to the disturbance in haplotypic structure and therefore, tried to filter AGs based on chunk counts from ChromoPainter results, preserving only AGs which are below the average chunk count of real genomes. The reasoning behind this was to preserve most realistic AGs with undisturbed

chunks. Even with this filtering, slight decrease in higher MAF bins was still present. Yet results of implementation with AGs for low frequency alleles and without AGs for high frequency ones could be combined to achieve the best performance. Although being not very practical in its current form, future improved models can become very useful, largely for GWAS studies in which imputation is a common application to increase resolution. Different generative models such as MaskGAN [16] which demonstrated good results in text gap filling might also be adapted for genetic imputation. RBM is possibly another option to be used as an imputation tool directly by itself, since once the weights have been learned, it is possible to fix a subset of the visible variables and to compute the average values of the unobserved ones by sampling the probability distribution (in fact, it is even easier than sampling entirely new configurations since the fixed subset of variables will accelerate the convergence of the sampling algorithm).

Scans for detecting selection are another promising use case for AGs as high-fidelity alternatives to real genomes. The XP-EHH and PBS scores computed on AGs were highly correlated with the scores of real genomes. In particular, the highest peak we obtained for Estonian genomes was also present in AGs, although it was the second highest peak in RBM XP-EHH plot (Fig 3). This peak falls within the range of skin color associated *SLC24A5* gene, which is potentially under positive selection in many European populations [52].

As an additional feature, training an RBM to model the data distribution gives access to a latent encoding of data points, providing a potentially easier to use representation of data (S15 Fig). Future works could enhance our current GAN model to also provide an encoding mechanism, in the spirit of [53–55]. It is expected that these interpretable representations of the data will be relevant for downstream tasks [54] and can be used as a starting point for various population genetics analyses.

We want to highlight that AGs are created without requiring the knowledge of the underlying evolutionary history, or the pre-processing bioinformatic pipelines (SNP ascertainment, data filtering). Unlike coalescent simulations, for which one needs to control parameters, AGs in their current form are solely constructed on raw information of real genomes. Our method offers a direct way to generate artificial genomes for any original dataset. On the other hand, the genomes generated using a coalescent simulator required substantial upstream work (from previous studies) as they were based on an explicitly parameterized model that had been inferred on real data using advanced methods for demographic reconstruction. In particular, this approach is not suitable when we want to generate AGs for highly complex datasets (eg full 1000 Genomes) for which it is arduous to infer a full evolutionary model accurately fitting the data and even more so, to mimic all the biases induced by potentially unknown bioinformatic pipelines. Last but not least, this coalescent generated data cannot be merged directly with real public genomes because there is no direct correspondence between the real SNPs and those generated, and coalescent approaches might struggle to match, among other things, real complex patterns of LD [35]. To summarize, while the classical coalescent simulator only allows unconditional sampling of a new haplotype $h$ from a predefined distribution $P(h|\theta)$ where the demographic parameters $\theta$ have to be given, our generative models learn how to generate $h$ from the conditional sampling distribution (CSD) $P(h|h_1,\dots h_n)$, where $(h_i)$ are the observed haplotypes. Computing, approximating or sampling from this CSD is known to be a difficult task [34,56,57].

We believe it will be possible in the future to extend our approach with conditional GAN/ RBM methods to allow fine control over the composition of artificial datasets based on (i) additional labels such as population names or any environmental covariate, or (ii) evolutionary parameters. While the former is based only on real datasets, the latter requires training on

genetic simulations (coalescent-based or forward) and has a different goal: it may provide an alternative simulator and/or permit inference of evolutionary models.

We envision three main applications of our generative methods: (i) improve the performance of genomic tasks such as imputation, ancestry reconstruction, GWAS studies, by augmenting public genomic panels with AGs that serve as proxies for private datasets that are not accessible; (ii) enable preliminary genomic analyses and proof-of-concept before committing to long term application protocols and/or to facilitate future collaborations to access private datasets; (iii) use the encoding of real data learned by generative models as a starting input of various tasks, such as recombination, demography or selection inference or yet unknown tasks.

Although there are currently some limitations, generative models will most likely become prominent for genetic research in the near future with many promising applications. In this work, we demonstrated the first possible implementations and use of AGs, particularly to be used as realistic surrogates of real genomes which can be accessed publicly without privacy concerns.

## Materials & methods

### Ethics statement

Genomes from Estonian Biobank were accessed with Approval Number 285/T-13 obtained on 17/09/2018 by the University of Tartu Ethics Committee.

### Data

We used 2504 individual genomes from 1000 Genomes Project (1000 Genomes Project Consortium 2015) and 1000 individuals from Estonian Biobank [26] to create artificial genomes (AGs). Additional 2000 Estonian genomes were used as a test dataset. Another Estonian dataset consisting of 8678 individuals which were not used in training were used for imputation via Impute2 software [32]. Analyses were applied to a highly differentiated 805 SNPs selected as a subset from [25], 3348 SNPs dispersed over the whole chromosome 15 and a dense 10000 SNP range/region from chromosome 15. In the data format we used, rows are individuals/haplotypes (instances) and columns are positions/SNPs (features). Each allele at each position is represented either by 0 or 1. In the case of phased data (haplotypes), each column is one position whereas in the case of unphased data, each two column corresponds to a single position with alleles from two chromosomes.

### GAN model

We implemented the GAN model using python-3.6, Keras 2.2.4 deep learning library with TensorFlow backend [58], pandas 0.23.4 [59] and numpy 1.16.4 [60]. We implemented a fully-connected generator network consisting of an input layer with the size of the latent vector size 600, one hidden layer with size proportional to the number of SNPs as SNP_number/1.2 rounded, another hidden layer with size proportional to the number of SNPs as SNP_number/1.1 rounded and an output layer with the size of the number of SNPs. The latent vector is drawn from a Gaussian distribution with zero-mean and unit-variance. The discriminator is also a fully-connected network including an input layer with the size of the number of SNPs, one hidden layer with size proportional to the number of SNPs as SNP_number/2 rounded, another hidden layer with size proportional to the number of SNPs as SNP_number/3 rounded and an output layer of size 1. All layer outputs except for output layers have LeakyReLU activation functions with leaky_alpha parameter 0.01 and L2 regularization parameter

0.0001. The generator output layer activation function is tanh and discriminator output layer activation function is sigmoid. Both discriminator and combined GAN were trained thanks to the Adam optimization algorithm with binary cross entropy loss function. We set the discriminator learning rate as 0.0008 and combined GAN learning rate as 0.0001. For 5000 SNP data, the discriminator learning rate was set to 0.00008 and the combined GAN learning rate was set to 0.00001. The training to test dataset ratio was 3:1. We used batch size of 32 and trained all datasets up to 20000 epochs. We also investigated stopping the training based on $AA_{TS}$ scores. The score was calculated at 200 epoch intervals. For 805 SNP data, $AA_{TS}$ converged very quickly close to optimum 0.5 score. However, the difference between $AA_{truth}$ and $AA_{syn}$ scores indicates possible overfitting to multiple data points so it was difficult to define a stopping point. For 10K SNP data, convergence was observed after ~30K epochs (to around 0.75) and reduced the number of fixed alleles in AGs but the gain was very minimal (S22 Fig). Additionally, GAN was prone to mode collapse especially after 20K epochs which resulted in multiple failed training attempts. Therefore, this study presents results for AGs generated at 20k epochs, since the first two PCs of AGs combined with real genomes were visually coherent for all targeted datasets (Fig 1 and S5 Fig). Note that it could be possible to utilize AGs before or after the 20K epoch point. During each batch in the training, when only the discriminator is trained, we applied smoothing to the real labels [1] by vectoral addition of random uniform distribution via numpy.random.uniform with lower bound 0 and upper bound 0.1. Elements of the generated outputs were rounded to 0 or 1. After the training is complete, it is possible to generate as many AGs as desired. The code is available at "https://gitlab.inria.fr/ml_genetics/public/artificial_genomes".

## RBM model

The RBM model consists of one visible layer of size $N_v$ and one hidden layer of size $N_h$ coupled by a weight matrix W. It is a probabilistic model of the joint distribution of visible $\{v_i, i = 1, \ldots N_v\}$ and hidden variables $\{h_j, j = 1, \ldots N_h\}$ of the form

$$P(v, h) = e^{-E(v,h)}$$

with

$$E(v, h) = \sum_{ij} W_{ij} v_i h_j + bias\ terms$$

Visible variables here are 0,1 as they represent reference/alternative alleles, while the hidden variable type depends on the chosen activation function (sigmoid or RELU). They are there to build dependencies among visible variables which by default are independent, via the interaction strength W. The weight matrix can be used in two different manners to interpret the learned model:

1. feature wise: for each hidden variable $j$ the vector $\{W_{ij}, i = 1, \ldots N_v\}$ represents a certain combination of SNPs which, if activated, will contribute to activate or inhibit this feature $j$. These features are expected to be characteristic of the data structure (such as the population structure) and the vector of feature activations should provide a suitable representation of individuals. If $N_v < N_h$ this corresponds to compressing the input representation.

2. direction wise: the SVD decomposition of W provides two sets of singular vectors with one corresponding to the visible space (visible axes) and the other one to the hidden representation (hidden axes). The vectors associated to the largest singular values offer the possibility to project the data in a low dimensional space. Dominant visible axes are expected to be similar to the principal component axes while dominant hidden axes are expected to produce more

separable datapoints due to non-linear activation mechanisms. We used the latter (i.e. the projection into the hidden space) to perform our non-linear dimension reduction of the 1000 Genomes data (see S13 Fig).

The RBM was coded in Julia [61], and all the algorithm for the training has been done by the authors. The part of the algorithm involving linear algebra used the standard package provided by Julia. Two versions of the RBM were considered. In both versions, the visible nodes were encoded using Bernoulli random variables {0,1}, and the size of the visible layer was the same size as the considered input. Two different types of hidden layers were considered. First with a sigmoid activation function (hence having discrete {0,1} hidden variables), second with ReLu (Rectified Linear unit) activations in which case the hidden variables were positive and continuous (they are distributed according to a truncated gaussian distribution when conditioning on the values of the visible variables). Results with sigmoid activation function were worse compared to ReLu so we used ReLu for all the analyses (S23 Fig). The number of hidden nodes considered for the experiment was Nh = 100 for the 805 SNP dataset and Nh = 500 for the 10k one. There is no canonical way of fixing the number of hidden nodes, in practice we checked that the number of eigenvalues learnt by the model was smaller than the number of hidden nodes, and that by adding more hidden nodes no improvement were observed during the learning. The learning in general is quite stable, in order to have a smooth learning curve, the learning rate was set between 0.001 and 0.0001 and we used batch size of 32. The negative term of the gradient of the likelihood function was approximated using the PCDk method [62], with k = 10 and 100 of persistent chains. As a stopping criterion, we looked at when the $AA_{TS}$ score converges to the ideal value of 0.5 when sampling the learned distribution. When dealing with large and sparse datasets for selection tests, RBM model did not manage to provide reasonable $AA_{TS}$ scores because the sampling is intrinsically difficult for large systems with strong correlation. In that case, we used visually coherent PCA results as a stopping criterion. Once the RBM is trained over the dataset, it is possible, in order to avoid running a very long Monte Carlo Markov Chain, to initialize the chain on the training set. However, in the case of the large dataset (Estonian), we observe that the RBM is overfitting the dataset and therefore, starting from the training dataset makes the overfitting even worse. In order to prevent this effect as much as possible, we used another independent dataset of Estonian individuals (denoted sampling set) to start the Monte Carlo Markov Chain. With this trick, we observe that the $AA_{TS}$ score exhibits less overfitting than when the Markov Chain was started on the training dataset. We measure the privacy scores for both training and sampling sets compared to a test set. Similar to the GAN, it is possible to generate as many AGs as wanted after training. The relevant RBM code is available at "https://gitlab.inria.fr/ml_genetics/public/artificial_ genomes".

## Bernoulli distribution model

We used python-3.6, pandas 0.23.4 and numpy 1.16.4 for the Bernoulli distribution model code. Each allele at a given position was randomly drawn given the derived allele frequency in the real population.

## Markov chain model

We used python-3.6, pandas 0.23.4 and numpy 1.16.4 for the Markov chain model code. For each generated sample alleles were drawn from left (position 0) to right. At the initial position the allele was set by drawing from a Bernoulli distribution parameterized with the real frequency. At a given position $p$ the allele $h_p$ was drawn in {0,1} according to its probability given the previous sequence window *of size w*, $P(h_p | h_{p-w}, \ldots, h_{p-1})$. This probability is computed

from the observed haplotype frequencies in real data. After the initial position, the sequence window size increased incrementally up to a predefined window size (5 or 10 SNPs). The relevant code is available at "https://gitlab.inria.fr/ml_genetics/public/artificial_genomes".

## HAPGEN2

We used HAPGEN2 [35] to generate our targeted region of chromosome 15 for as many individuals as in the original dataset. We provided the training dataset (e.g. either 1000 Genomes or Estonian) and a recombination map [37] of the region as input. We sampled only control individuals and no cases. All other parameters were set to default.

## Coalescent simulations

We used stdpopsim [38] with the command line "stdpopsim HomSap -c chr15 -o CEU_chr15. trees -g HapMapII_GRCh37 -d OutOfAfrica_3G09 0 2000 0" to generate 2000 CEU haplotypes based on the demographic parameters inferred by Gutenkunst et al. 2009 [36]. We then selected the genome region corresponding to one targeted when generating AGs.

## Summary statistics

We define here the statistics that are not commonly used in population genetics. The 3-point scores measure the correlation patterns for SNP triplets. The 3-point correlation for SNPs $i$, $j$, and $k$ is defined as [63]:

$$c_{ijk}(a,b,c) = f_{ijk}(a,b,c) - f_{ij}(a,b)f_k(c) - f_{ik}(a,c)f_j(b) - f_{jk}(b,c)f_i(a) + 2f_i(a)f_j(b)f_k(c),$$

where the alleles $(a,b,c) \in \{0,1\}^3$, $f_i(a)$ is the frequency of allele $a$ at SNP $i$, $f_{ij}(a,b)$ is the frequency of the combination of allele $a$ at SNP $i$ and $b$ at SNP $j$, and finally $f_{ijk}(a,b,c)$ is the frequency of the combination $(a,b,c)$ at SNPs $(i,j,k)$. We computed the 3-point correlations for 8,000 randomly-picked triplets under different conditions (SNPs separated by 1, 4, 16, 64, 256, 512 or 1024 SNPs, as well as SNPs chosen at random) in each dataset.

PCA were computed on all datasets combined (e.g. Fig 1) as well as on "pairs" of datasets (the combination of real and a single type of generated data). 2D-Wasserstein distances for these paired PCA representations were computed based on the entropic regularized optimal transport problem with square euclidean distances computed from PCs 1 and 2 and a regularization parameter set to 0.001 (POT library, [64]).

To have reference values regarding the best achievable distances or correlations between real and generated summary statistics, we split randomly the 1000Genomes dataset in two and considered half of it as the real dataset and half as a "perfectly generated" dataset (called test).

## Chromosome painting

We compared the haplotype sharing distribution between real and artificial chromosomes through ChromoPainter [27]. In detail, we have painted 100 randomly selected "real" and "artificial" Estonians (recipients) against all the 1000 Genome Project phased data (donors). The nuisance parameters -n (348.57) and -M (0.00027) were estimated running 10 iterations of the expectation-maximization algorithm on a subset of 3,800 donor haplotypes.

## Haplostrips

We used Haplostrips [28] to visualize the haplotype structure of real and artificial genomes. We extracted 500 individuals from each sample set (Real, GAN AGs, RBM AGs) and considered them as different populations. Black dots represent derived alleles, white dots represent

ancestral alleles. The plotted SNPs were filtered for a population specific minor allele frequency >5%; haplotypes were clustered and sorted for distance against the consensus haplotype from the real set. See the application article for further details about the method.

## Nearest neighbour adversarial accuracy ($AA_{TS}$) and privacy loss

We used the following equations for calculating $AA_{TS}$ and privacy loss scores [29]:

$$AA_{truth} = \frac{1}{n} \sum_{i=1}^{n} \mathbf{1}(d_{TS}(i) > d_{TT}(i))$$

$$AA_{syn} = \frac{1}{n} \sum_{i=1}^{n} \mathbf{1}(d_{ST}(i) > d_{SS}(i))$$

$$AA_{TS} = \frac{1}{2} \left( AA_{truth} + AA_{syn} \right)$$

$$Privacy\ Loss = Test\ AA_{TS} - Train\ AA_{TS}$$

where n is the number of real samples as well as of artificial samples; 1 is a function which takes the value 1 if the argument is true and 0 if the argument is false; $d_{TS}(i)$ is the distance between the real genome indexed by i and its nearest neighbour in the artificial genome dataset; $d_{ST}(i)$ is the distance between the artificial genome indexed by i and its nearest neighbour in the real genome dataset; $d_{TT}(i)$ is the distance of the real genome indexed by i to its nearest neighbour in the real genome dataset; $d_{SS}(i)$ is the distance of the artificial genome indexed by i to its nearest neighbour in the artificial genome dataset. An $AA_{TS}$ score of 0.5 is optimal whereas lower values indicate overfitting and higher values indicate underfitting. For a better resolution for the detection of overfitting, we also provided $AA_{truth}$ and $AA_{syn}$ metrics identified in the general equation of $AA_{TS}$. If $AA_{TS}$ is 0.5 but $AA_{truth}$ 0 and $AA_{syn}$ 1, this means that the model is not overfitting in terms of a single data point but multiple ones. In other words, the model might be focusing on small batches of similar real genomes to create artificial genomes clustered at the center of each batch. Privacy loss is the difference of $AA_{TS}$ score of AGs calculated against the training samples set and a different real test set which was not used in training.

## Selection tests

We used scikit-allel package for XP-EHH [65] and PBS [66] tests. We used 1000 Estonian individuals (2000 haplotypes) with 3348 SNPs which were homogenously dispersed over chromosome 15 (spanning the whole chromosome with similar distance between SNPs) for the training of GAN and RBM models. For XP-EHH, Yoruban (YRI, 216 haplotypes) population from 1000 Genomes data was used as the complementary population. For PBS, Yoruban (YRI, 216 haplotypes) and Japanese (JPT, 208 haplotypes) populations from 1000 Genomes data were used as complementary populations. PBS window size was 10 and step size was 5, resulting in 668 windows. 216 real and 216 AG haplotypes were compared for the analyses.

## Supporting information

**S1 Fig. Generative adversarial network (GAN) scheme.**
(TIF)

**S2 Fig. Restricted Boltzmann machine (RBM) scheme.**
(TIF)

**S3 Fig.** Uniform manifold approximation and projection (UMAP) of real genomes from 1000 Genomes data spanning 805 SNPs along with artificial genome counterparts created via **a)** Bernoulli, **b)** Markov chain (with 10 window length), **c)** GAN and **d)** RBM models.
(TIF)

**S4 Fig. Distribution of haplotypic pairwise difference within (left) and between (right) datasets of real genomes from 1000 Genomes data spanning 805 SNPs and artificial genome counterparts generated using different models.**
(TIF)

**S5 Fig.** PCA of real genomes (gray) from **a)** 1000 Genomes data and **b)** Estonian Biobank spanning 10K SNPs along with artificial genome counterparts generated using GAN (blue) and RBM (red) models.
(TIF)

**S6 Fig.** Distribution of haplotypic pairwise difference within (left) and between (right) datasets of real genomes from **a)** 1000 Genomes data and **b)** Estonian Biobank spanning 10K SNPs and artificial genome counterparts generated using GAN and RBM models.
(TIF)

**S7 Fig.** Allele frequency comparison of corresponding SNPs between real genomes from Estonian Biobank spanning 10K SNPs and artificial genome counterparts generated using GAN and RBM models as **a)** the whole range and **b)** zoomed to low frequencies. Clustering below the diagonal in the low frequency section for the GAN plot indicates insufficient representation of rare alleles in artificial genomes.
(TIF)

**S8 Fig.** Distribution of minimum distance to the closest neighbour for real genomes from **a)** 1000 Genomes data and **b)** Estonian Biobank spanning 10K SNPs along with artificial genome counterparts generated via GAN and RBM models.
(TIF)

**S9 Fig. LD comparison of real (Estonian) vs generated datasets.**
(TIF)

**S10 Fig. 3-point correlation statistics for SNPs separated by different distances.**
(TIF)

**S11 Fig. Haplostrips showing the mixed nature of haplotype structures for real Estonian (gray rows) along with GAN (blue rows) and RBM (red rows) haplotypes.**
(TIF)

**S12 Fig.** Chromosome painting of two **a)** real Estonian genomes, **b)** GAN and **c)** RBM artificial Estonian genomes with 1000 Genomes donors colored based on super population codes. EUR–European, EAS–East Asian, AMR–Admixed American, SAS–South Asian, AFR–African.
(TIF)

**S13 Fig. a)** Nearest neighbour adversarial accuracy (AA$_{TS}$) scores of artificial genomes generated from Estonian Biobank. Black line indicates the optimum value whereas values below the line indicate overfitting and values above the line indicate underfitting. **b)** Privacy loss. Test1 is

a separate set of real Estonian genomes. Positive values indicate information leakage, hence overfitting.
(TIF)

**S14 Fig. $AA_{TS}$ and privacy loss change of RBM AGs over epochs.**
(TIF)

**S15 Fig. Comparison of PCA (right column) and non-linear dimension reduction via RBM (left column) for real genomes from 1000 Genomes data spanning 805 SNPs.** The RBM reduction was obtained by projecting the real data into the hidden space of the RBM (see Materials & Methods). Population codes are as defined by the 1000 Genomes Project.
(TIF)

**S16 Fig. Activations of each of the 100 nodes belonging to the RBM hidden layer when applied to the real genomes from 1000 Genomes data spanning 805 SNPs.** For each hidden node the X-axis corresponds to the real haplotypes and Y-axis to the activation of the node by a single haplotype. On the X-axis, haplotypes are ordered by region (Africa, America, East Asia, European, East Asia) and colored by population. Because this RBM activation function is a ReLU with threshold 0 (by design), all values are positive and a zero-value indicates that the node is not activated by a given haplotype. The ordering of nodes has no specific meaning.
(TIF)

**S17 Fig.** Analyses of artificial genomes generated by HAPGEN2 showing **a)** PCA of generated (green) performed with real Estonian genomes (grey) and **b)** distribution of minimum distance to the closest neighbour displaying real Estonian genomes (grey), HAPGEN2 (green), GAN (blue) and RBM (red) artificial genomes.
(TIF)

**S18 Fig. a)** PCA of real (Estonian) and artificial genomes simulated via coalescent approach using stdpopsim (CEU). **b)** Allele frequency quantiles of real (Estonian) vs artificial genomes simulated via coalescent approach using stdpopsim (CEU).
(TIF)

**S19 Fig. a)** LD as a function of SNP distance after removing sites that are fixed in at least one dataset and removing alleles below 0.1 frequency from all datasets. Pairwise SNP distances were stratified into 50 bins and for each distance bin, the correlation was averaged over all pairs of SNPs belonging to the bin. Allele frequency quantiles of real (Estonian) vs **b)** GAN Estonian artificial genomes, **c)** RBM Estonian artificial genomes and **d)** artificial genomes simulated via coalescent approach using stdpopsim (CEU).
(TIF)

**S20 Fig. Comparison of sites which are polymorphic in real genomes from Estonian Biobank but fixed in artificial genome counterparts generated via GAN and RBM models.**
(TIF)

**S21 Fig.** Sensitivity tests for **a)** $AA_{TS}$ (scores over 0.5 indicate underfitting and below 0.5 indicate overfitting) and **b)** privacy scores (orange and red lines to mark the difference between RBM trained up to 350 and 690 epochs). All datasets consist of 2000 samples. Test1 and Test2 are real Estonian individuals who were not used in training. Mixed1 dataset has 1 real individual from the training dataset, Mixed2 has 10, Mixed3 has 50, Mixed4 has 100, Mixed5 has 500 and Mixed6 has 1000 individuals.
(TIF)

**S22 Fig.** Evaluation of $AA_{TS}$ scores of the GAN model for artificial Estonian genomes spanning **a)** 805 highly informative SNPs and **b)** dense 10K SNPs along with the total fixed sites for the outputs of epochs at 200 intervals.
(TIF)

**S23 Fig.** Comparison of **a)** $AA_{TS}$ score and **b)** linkage disequilibrium of artificial genomes created via RBM model with sigmoid and ReLu activation functions.
(TIF)

**S1 Table. Genotype/Phenotype contingency table for real and artificial Estonian genomes (AG).** Ancestral allele "A" is associated with brown eye color and derived allele "G" is associated with blue eye color phenotype.
(DOCX)

**S1 Text. Preliminary analysis on generating artificial genomes with corresponding phenotypes.**
(DOCX)

## Acknowledgments

Thanks to Inria TAU team and High Performance Computing Center of the University of Tartu for providing computational resources. Thanks to Isabelle Guyon and Adrien Pavao for their valuable insight into $AA_{TS}$ score, Susana Ribeiro for comments on the manuscript and Lofti Slim for discussion.

## Author Contributions

**Conceptualization:** Burak Yelmen, Aurélien Decelle, Luca Pagani, Flora Jay.

**Data curation:** Burak Yelmen, Flora Jay.

**Formal analysis:** Burak Yelmen, Aurélien Decelle, Linda Ongaro, Davide Marnetto, Francesco Montinaro, Cyril Furtlehner, Flora Jay.

**Funding acquisition:** Luca Pagani, Flora Jay.

**Investigation:** Burak Yelmen, Aurélien Decelle, Flora Jay.

**Methodology:** Burak Yelmen, Aurélien Decelle, Cyril Furtlehner, Luca Pagani, Flora Jay.

**Project administration:** Burak Yelmen, Luca Pagani, Flora Jay.

**Resources:** Burak Yelmen, Luca Pagani, Flora Jay.

**Software:** Burak Yelmen, Aurélien Decelle, Flora Jay.

**Supervision:** Luca Pagani, Flora Jay.

**Visualization:** Burak Yelmen, Aurélien Decelle, Linda Ongaro, Flora Jay.

**Writing – original draft:** Burak Yelmen.

**Writing – review & editing:** Burak Yelmen, Aurélien Decelle, Linda Ongaro, Davide Marnetto, Corentin Tallec, Francesco Montinaro, Cyril Furtlehner, Luca Pagani, Flora Jay.

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
