## [Decision Letter · Decision Letter 0]

5 Nov 2020

Dear Dr Yelmen,

Thank you very much for submitting your Research Article entitled 'Creating Artificial Human Genomes Using Generative Neural Networks' to PLOS Genetics. Your manuscript was fully evaluated at the editorial level and by independent peer reviewers. The reviewers appreciated the attention to an important topic but identified some aspects of the manuscript that should be improved.

We therefore ask you to modify the manuscript according to the review recommendations before we can consider your manuscript for acceptance. Your revisions should address the specific points made by each reviewer.

[LINK]

Yours sincerely,

Sara Mathieson

Guest Editor

PLOS Genetics

Hua Tang

Section Editor: Natural Variation

PLOS Genetics

As the reviewers note, the authors have done a thorough job of addressing the previous comments, but a few issues remain. These issues should still be addressed in full, including that the phenotype-genotype application be omitted or addressed only as future work.

Reviewer's Responses to Questions

**Comments to the Authors:**

Reviewer #1: I reviewed a previous submission of this manuscript, and the authors have addressed essentially all of my prior comments. In particular, the exposition and the discussion section are greatly improved.

I have a few minor comments that I list below.

-- In discussion of using the coalescent as a generative model it may be helpful to discuss that the GAN is essentially learning something akin to the posterior predictive distribution: P(X | X_1, ..., X_N, \\Theta) for observed haplotypes X_1,...,X_N and demographic parameters \\Theta. Such conditional probabilities are notoriously hard to compute or even sample from under the coalescent, spawning a number of approximations (e.g., Li and Stephens' copying model; and Josh Paul's series of papers on Conditional Sampling Distributions), further highlighting the utility of the present method. The usual coalescent simulations, by contrast, are merely simulating under the prior, P(X | \\Theta). Adding something to this effect in the discussion may help clarify some of the novelty and benefit of the present method, and clear up any confusion on why additional haplotypes cannot easily be simulated from the coalescent even if the demographic model is known.

-- Mode collapse during GAN training has--to some extent--been solved by adding an entropic penalty (Prescribed Generative Adversarial Networks by Dieng, Ruiz, Blei, and Titsias). It is certainly not necessary to incorporate this into the present manuscript, but may be of future use to the authors.

-- Lines 325-328 are unclear and difficult to understand.

-- Line 378: The "this" in "we initially speculated that this" is unclear. I believe it's related to the decrease in performance for higher frequency alleles, but it currently reads like it's referring to the fact that increasing the number of AGs did not affect the results.

Reviewer #2: The authors have done a nice enough job on the revision, and I'm happy with most of it however I have a few sticking points still.

1) the comparison between simulated CEU and Estonians seems undesirable-- the authors are using 1000 Genomes data -- why not do a direct comparison to the CEU genomes in that dataset?

2) again i find the phenotypic analysis to be too cursory for publication.

Reviewer #3: This is my review of the revised submission of "Creating Artificial Human Genomes Using Generative Neural Networks". My primary concern with the initial submission was the insufficient discussion of the practical utility of the authors' approach to generating artificial genomes. This has been largely addressed in the revision but I do have several outstanding requests:

1. The authors have done little to clarify the utility of recapitulating in AGs the phenotype-genotype associations from real data. What could these be used for? Sharing more thoughts on this could help readers see the value of the authors' approach.

2. Please elaborate on the method's ability to create larger samples. The authors response on this issue seems to imply that multiple independent runs can be combined to create a sample with a larger number of genomes. In this case, our hope would be that the resulting artificial population sample would to some degree makes sense from an evolutionary standpoint, i.e. the genomes in that sample all implicitly belong to an ancestral recombination graph that is more or less drawn from the same distribution as that from which the GAN was trained. But if we take two AG samples and combine them to make a larger sample, will this be the case, or will they look like two different populations (or in the case of gene-scale AGs, two different loci) that were stitched together? I may be misunderstanding, but I do not think that the authors' response and revision addresses this at all. One way in which this could be tested is by making sure that FST between the two AG samples that were combined together is not elevated.

3. I reiterate my request to include the Supplementary Text on RBMs in the main text. The authors' response about size limitations is not valid. Per the PLoS Gen website: "Manuscripts can be any length. There are no restrictions on word count, number of figures, or amount of supporting information."

4. Regarding the legend for Supplementary Figure 7: please more explicitly state that this is a scatter plot comparing real and artificial allele frequencies between ANALOGOUS polymorphisms in the two datasets.

5. The authors' new results examining the RBM scheme that has reduced privacy loss are interesting. They should be included in the paper rather than simply stating that this RBM "had only a slight decrease in quality of the summary statistics" while sharing no data.

6. The precise stopping criteria for GAN training were not added to the text as requested in my original review. Please rectify this.

**Have all data underlying the figures and results presented in the manuscript been provided?**

Reviewer #1: Yes

Reviewer #2: Yes

Reviewer #3: None

PLOS authors have the option to publish the peer review history of their article (what does this mean?). If published, this will include your full peer review and any attached files.

Reviewer #1: **Yes: **Jeffrey P. Spence

Reviewer #2: No

Reviewer #3: No

---

## [Editor Report · Decision Letter 1]

8 Dec 2020

Dear Dr Yelmen,

We are pleased to inform you that your manuscript entitled "Creating Artificial Human Genomes Using Generative Neural Networks" has been editorially accepted for publication in PLOS Genetics. Congratulations!

Yours sincerely,

Sara Mathieson

Guest Editor

PLOS Genetics

Hua Tang

Section Editor: Natural Variation

PLOS Genetics

Comments from the reviewers (if applicable):

**Data Deposition**

http://datadryad.org/submit?journalID=pgenetics&manu=PGENETICS-D-20-01456R1

**Press Queries**

---

## [Editor Report · Acceptance letter]

15 Jan 2021

PGENETICS-D-20-01456R1 

Creating Artificial Human Genomes Using Generative Neural Networks 

Dear Dr Yelmen, 

We are pleased to inform you that your manuscript entitled "Creating Artificial Human Genomes Using Generative Neural Networks" has been formally accepted for publication in PLOS Genetics! Your manuscript is now with our production department and you will be notified of the publication date in due course.

With kind regards,

Melanie Wincott

PLOS Genetics

On behalf of:
